# Axon initial segment geometry in relation to motoneuron excitability

**Travis M. Rotterman**[1]*, **Darío I. Carrasco**[1], **Stephen N. Housley**[1], **Paul Nardelli**[1], **Randall K. Powers**[2], **Timothy C. Cope**[1,3]*

**1** School of Biological Sciences, Georgia Institute of Technology, Atlanta, GA, United States of America,
**2** Department of Physiology and Biophysics, University of Washington, Seattle, WA, United States of
America, **3** W.H. Coulter Department of Biomedical Engineering, Emory University and Georgia Institute of
Technology, Atlanta, GA, United States of America

\* trotterman3@gatech.edu (TMR); tim.cope@gatech.edu (TCC)

**Data Availability Statement:** All data and code used in this paper are currently publicly available: https://github.com/nickh89/plos_one_2021_rotterman.

## Abstract

The axon initial segment (AIS) responsible for action potential initiation is a dynamic structure that varies and changes together with neuronal excitability. Like other neuron types, alpha motoneurons in the mammalian spinal cord express heterogeneity and plasticity in AIS geometry, including length ($AIS_l$) and distance from soma ($AIS_d$). The present study aimed to establish the relationship of AIS geometry with a measure of intrinsic excitability, rheobase current, that varies by 20-fold or more among normal motoneurons. We began by determining whether AIS length or distance differed for motoneurons in motor pools that exhibit different activity profiles. Motoneurons sampled from the medial gastrocnemius (MG) motor pool exhibited values for average $AIS_d$ that were significantly greater than that for motoneurons from the soleus (SOL) motor pool, which is more readily recruited in low-level activities. Next, we tested whether $AIS_d$ covaried with intrinsic excitability of individual motoneurons. In anesthetized rats, we measured rheobase current intracellularly from MG motoneurons *in vivo* before labeling them for immunohistochemical study of AIS structure. For 16 motoneurons sampled from the MG motor pool, this combinatory approach revealed that $AIS_d$, but not $AIS_l$, was significantly related to rheobase, as AIS tended to be located further from the soma on motoneurons that were less excitable. Although a causal relation with excitability seems unlikely, $AIS_d$ falls among a constellation of properties related to the recruitability of motor units and their parent motoneurons.

## Introduction

The axon initial segment (AIS) responsible for action potential initiation in vertebrate neurons is a dynamic structure [1–3]. From its position on the proximal axon, AIS geometry adjusts variously in length ($AIS_l$) and in distance from the soma ($AIS_d$) during development and in response to changes in neural activity [4–6]. AIS geometry is also found to vary in relation to diverse firing behaviors of neurons belonging to the same or different populations of neuron types [7–13]. For example, $AIS_d$ and $AIS_l$ vary in relation to the different sound-frequency sensitivities of neurons in avian auditory nuclei [14]. While ongoing experimental and

**Funding:** 1) TC, R01CA221363, National Institutes of Health: National Cancer Institute (https://www.cancer.gov/), The funders had no role in study design, data collection and analysis, decision to publish, or preparation of the manuscript. 2) TR, F32NS112556, National Institutes of Health: National Institute Neurological Disorders and Stroke (https://www.ninds.nih.gov/), The funders had no role in study design, data collection and analysis, decision to publish, or preparation of the manuscript.

**Competing interests:** The authors have declared that no competing interests exist.

computational studies advance biophysical explanations for the influence of AIS geometry on neuronal excitability [4–6,15,16], investigation remains limited to relatively few neuron types. The present study aimed to extend examination of AIS geometry and its direct relation to excitability of spinal motoneurons *in vivo*.

Recent studies demonstrate modification of AIS geometry in mouse motoneurons under pathological conditions that reduce motor activity. Muscle injection with botulinum toxin known to induce compensatory increases in motoneuron excitability also increases both AIS length and distance [17]. Although this finding provides indirect evidence that AIS geometry and excitability covary in motoneurons, a mouse model of motoneuron disease yields changes in AIS geometry that are unaccompanied by any detectable modification of excitability [18,19]. While these studies provide clear evidence of the dynamic nature of AIS geometry, they leave uncertainty about the relationship with motoneuronal excitability, especially given the complex effects that pathological conditions have on multiple properties that affect excitability [20–25]. Uncertainty also arises, because AIS geometry and excitability were not measured together in individual motoneurons. We turned our attention instead to untreated control rodents with interest in the untested possibility that the substantial variability in AIS geometry reported in those and one earlier study [26] might relate to the wide diversity in the native excitability of motoneurons in normal adult animals.

Ample evidence reveals clear differences in the excitability of spinal motoneurons [27–29]. Differences are readily observed in the typical recruitment sequence of motor units made up of subsets of skeletal muscle fibers, each directly controlled by a motoneuron. During graded muscle contractions, individual motor units are successively recruited as contraction builds over a wide range of whole muscle force [28]. The emergent recruitment order of motor units relies on the scaling of the intrinsic excitability of motoneurons, which spans up to a 30-fold range when measured as rheobase current, defined as the minimum steady current required to reach firing threshold [30]. Characteristic differences in muscle activity patterns also depend on the relative excitability among motoneurons comprising the motor pools supplying individual muscles. The broad range in excitability of motoneurons and motor pools affords an advantageous condition for testing possible covariation with AIS geometry.

The present study aimed to test the hypothesis that AIS geometry covaries with the endogenous excitability of motoneurons in healthy adult rats. In a first of two experimental approaches, we examined AIS geometry of motoneurons in motor pools having known characteristic differences in activity patterns. Next, we directly evaluated the relationship between AIS geometry and intrinsic excitability measured *in vivo* from individual motoneurons. Results of both approaches supported our hypothesis and were consistent in showing that $AIS_d$ varied inversely with the native excitability of spinal motoneurons, i.e., AIS was located more distally from the soma for less excitable motoneurons. Present findings associate AIS geometry with a measure of motoneuron excitability that aligns with the orderly recruitment of motor units. In our discussion we conclude that while it is unlikely to have a strong causal role in determining motoneuron excitability, AIS geometry falls among a constellation of coordinated motor unit properties related to recruitment order.

## Results

### $AIS_d$ differs in motor pools exhibiting different activity levels

Motoneurons comprising the soleus (SOL) and medial gastrocnemius (MG) motor pools were examined for differences in $AIS_l$ and $AIS_d$. These motor pools were selected for their contrasting activity patterns observed in animals engaged in movement behavior [31–36]. Motor pool activity monitored by EMG or force production of the associated muscle is prominent in SOL

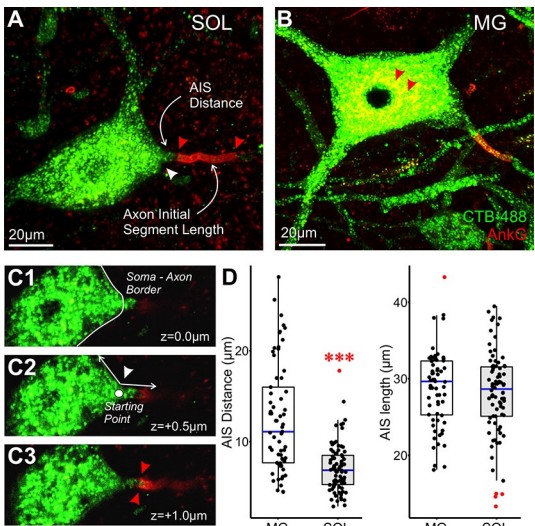

**Fig 1. AIS distance differs in motor pools distinguished by their activity patterns.** (A, C) MG and SOL motoneurons pre-labeled with the retrograde tracer cholera toxin subunit B conjugated to Alexa fluor 488 (CTB) and immunolabeled for ankyrin-G (AnkG), a marker for the AIS. Our definition of AIS distance ($AIS_d$) and AIS length ($AIS_l$) is identified by white arrows and corresponding text in panel C. (A) white arrow indicates the start of axon from soma and first red arrow indicates initiation of AIS. The two red arrowheads identify the start and end of the AIS. The two red arrowheads in C point to lipofuscin granules that is common in mature motoneurons. (B1-B3) Single z-steps through a confocal image stack. (B1) Initial indication of the axon emerging from the soma. The soma-axon boundary is indicated by the solid white line. (B2) The white lines were used to determine the apex of the angle in which the axon emerged from the soma and is marked by a white dot. This was used as the starting point for measuring distance. (B3) The red arrows indicate the start of the AIS as defined by expression of AnkG. (D) Box and whisker plots of $AIS_d$ and $AIS_l$ compared between MG (n = 65) and SOL (n = 82) motoneurons. Blue horizontal line inside the plot represents the median value. Black dots represent a single data point and red dots are designated outliers. Based on the distribution of these data, a nonparametric test (Mann-Whitney U test) was used for an initial statistical comparison ($AIS_d$: p <0.001, $AIS_l$: p = 0.38).

during quiet standing and slow locomotor speed, while activity in MG develops progressively with movement intensity, approaching maximum only during rapid and more vigorous movements [37] The different activity patterns reflect different proportions along the spectrum of intrinsic excitability for motoneurons, with highly excitable motoneurons dominating the SOL motor pool and a preponderance lower excitability motoneurons populating the MG pool [38–41]. We examined these two motor pools for corresponding differences in AIS geometry.

Motoneurons belonging to SOL or MG motor pools were identified by retrograde CTB labeling in 3-dimensional spinal cord cross sectional images obtained from 5 rats. Unequivocal visualization of the full length and position of the AIS delineated by expression of ankyrin G (AnkG) reactivity [42,43] was obtained for 82 SOL and 65 MG motoneurons (Fig 1A–1C). On the whole, values for $AIS_l$ and $AIS_d$ reported in Table 1 covered ranges similar to those provided in the published reports on motoneurons, which were obtained from mice and rats [17,18,26]. $AIS_l$ was not significantly different for motoneurons sampled the two motor pools (Table 1). However, differentiating motoneurons by motor pool membership exposed a significant difference in $AIS_d$ represented in comparison of images obtained from two motoneurons, one each from the SOL (Fig 1A) and MG (Fig 1C) motor pool. While the sample distributions overlapped considerably over short distances (<10 μm), very few SOL motoneurons had $AIS_d$ values extending beyond the median value (12.50 μm) observed for the sample of MG motoneurons (Figs 1D and S1). Motor pool differences were probably not the result of $AIS_d$ scaling with motoneuron soma size, at least as measured by maximum cross-sectional

**Table 1. Population data measurements from retrogradely labeled MG and SOL motoneurons.**

| Population Summary | | | | |
|---|---|---|---|---|
| *MG (n = 65)* | *Mean* | *Std. Dev.* | *Min* | *Max* |
| **AIS Distance ($AIS_d$)** | 12.50 | 5.95 | 4.48 | 28.10 |
| **AIS Length ($AIS_l$)** | 28.78 | 4.96 | 18.10 | 43.30 |
| **Soma Diameter** | 37.14 | 5.15 | 26.55 | 48.50 |
| *SOL (n = 82)* | | | | |
| **AIS Distance ($AIS_d$)** | 7.11 | 2.65 | 2.88 | 17.80 |
| **AIS Length ($AIS_l$)** | 28.00 | 5.62 | 13.30 | 39.50 |
| **Soma Diameter** | 36.47 | 5.37 | 22.40 | 51.20 |

diameter, since soma diameter was not significantly different in the SOL and MG pools (Table 1).

## AIS geometry in relation to motoneuron morphology and physiology

Results described above established differences in $AIS_d$ between SOL and MG motor pools. This distinction may be linked to characteristic differences in motor pool activity levels and/or in motoneuron excitability in these two motor pools. Given associations with neuronal excitability found in experimentally other systems (see Goethals and Brette 2020), we hypothesized that AIS geometry covaries with motoneuron excitability and related properties. We tested this hypothesis for individual motoneurons studied electrophysiologically *in vivo* and labeled for morphological analyses. Motoneurons from the MG motor pool were selected for study, because they cover the full range of excitability exhibited by all spinal motoneurons [44–46]. Restricting assessments to motoneurons from a single motor pool eliminated possible confounding influences introduced by unmeasured or unknown differences between motor pools.

Intracellular recording and labeling of 18 individual motoneurons *in vivo* enabled matching their morphological and electrophysiological properties. Sample data are shown in Fig 2. Images in Fig 2A represent morphology of the majority of motoneurons (16/18) for which the AIS-bearing axon clearly emerged directly from the soma. Axons originated from dendrites in the remaining minority of cases (11%; Fig 2C), comparable to the small percent found in mouse motoneurons [26]. Fig 2B shows physiological responses conventionally used to characterize motoneurons [38]. Additional properties were measured when stable recording conditions permitted. Input conductance was assessed as a contributor to rheobase, while action potential after-hyperpolarization duration and motor unit twitch evoked by the parent motoneuron provided additional representation of the motoneuron's functional specialization.

Values for multiple morphological properties from 3-D confocal image stacks together with various electrophysiological properties are compiled from the 16 remaining MG motoneurons in Tables 2 and 3. All properties were comparable to those that have been measured in previous studies of rodent motoneurons [44,47–49]. Particularly important to our study goal, the sample represents a substantial portion of the reported distribution for rheobase among MG motoneurons [44,45]. Also pertinent was the range in $AIS_d$ for this sample of MG motoneurons, which at 27μm covers the span associated with meaningful differences in excitability of other neuron types.

In support of our hypothesis, Fig 3A shows that $AIS_d$ covaried with differences in excitability among motoneurons in normal, mature animals. Specifically, motoneurons with relatively short $AIS_d$ tended to have lower rheobase, i.e., higher excitability than motoneurons with longer $AIS_d$. Differences in $AIS_d$ accounted for 63.1% of the variance ($R^2$) in rheobase. This

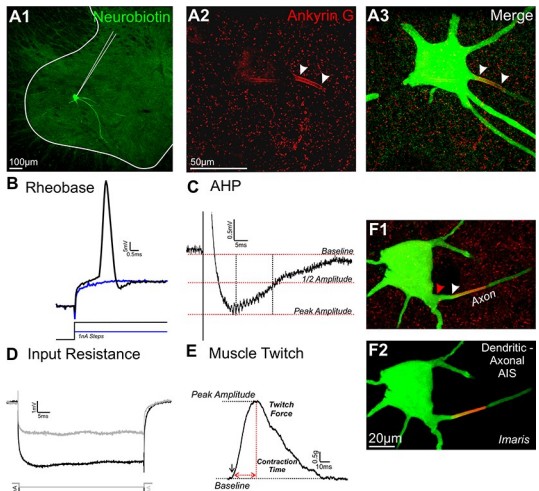

**Fig 2. Morphological and *in vivo* electrophysiological parameters measured in individual Neurobiotin filled motoneurons.** (A1) Collapsed confocal image (10x) of a Neurobiotin filled motoneuron in the ventral horn of the spinal cord. Solid white line indicates white/gray matter boundary. (A2-3) High magnification confocal image of AnkG immunoreactivity in ventral horn of the spinal cord (A1). (A3) Image of Neurobiotin filled MG motoneuron revealed with streptavidin-488 merged with AnkG. White arrow heads indicate AIS. (B) Rheobase generated by 50ms square pulse of depolarizing current injection. Blue trace indicates a current injection that did not elicit an action potential. (C) After hyperpolarization (ahp) was elicited via 0.5ms suprathreshold current injection. (D) Input resistance measured by hyperpolarizing currents (-1/3nA) for 50ms. Input conductance is reported as the inverse of resistance. (E) Muscle twitch results from ahp current injection. Twitch force is defined as the maximum amplitude (vertical dashed red line). Contraction time (CT) is measured between the twitch onset to the maximum amplitude (horizontal dashed red line). (F1-2) A small percent (11% in our sample) of axons are of dendritic origin. (F1) Neurobiotin filled motoneuron with a parent dendrite indicated with the red arrowhead and the protruding axon indicated with the white arrowhead. (F2) Merge Imaris rendition of image displayed in F1.

finding is qualitatively consistent with comparisons made above for SOL and MG motor pools in that the distribution of $AIS_d$ shifted toward shorter values in more readily active SOL motor pool (Fig 1). Also consistent was the finding that $AIS_l$ was not significantly correlated with either motoneuron rheobase ($R^2 = 0.161$) or $AIS_d$ ($R^2 = 0.033$).

The two motoneurons for which axons originated from dendrites are excluded from Fig 3A. The small sample size precludes further evaluation, other than to note similarity with hippocampal neurons for which excitability is disproportionally greater when axons originate from dendrites as opposed to soma [7–9].

## AIS geometry correlations with other motoneuron properties

Rheobase reflects determinants of excitability residing not only at the AIS, but also in the soma-dendritic compartment of the motoneuron [15]. Earlier studies of spinal motoneurons

**Table 2. Morphological characteristics of motoneurons and the AIS.**

| Morphological Measurements | | | | | | |
|---|---|---|---|---|---|---|
| *Variable* | *Mean* | *Std. Dev.* | *SE* | *N* | *Min* | *Max* |
| **AIS Distance (μm)** | 13.38 | 7.92 | 1.98 | 16 | 5.58 | 32.61 |
| **Prox. Axon Surface (μm²)** | 245.74 | 163.96 | 40.99 | 16 | 88.22 | 617.65 |
| **AIS Length (μm)** | 26.98 | 5.00 | 1.25 | 16 | 18.23 | 36.98 |
| **AIS Surface (μm²)** | 245.25 | 78.03 | 19.51 | 16 | 123.76 | 394.71 |
| **Soma Surface (μm²)** | 5026.42 | 861.29 | 215.32 | 16 | 3391.10 | 6687.74 |
| **Soma Volume (μm³)** | 23305.44 | 4361.25 | 1090.31 | 16 | 13139.70 | 30398.90 |
| **Max Soma Diam. (μm)** | 54.15 | 7.50 | 1.94 | 15 | 41.65 | 70.19 |

**Table 3. Biophysical measurements from intracellularly recorded MG motoneurons.**

| Conventional Motoneuron and Motor Unit Measurements | | | | | | |
|---|---|---|---|---|---|---|
| *Variable* | *Mean* | *Std. Dev.* | *SE* | *N* | *Min* | *Max* |
| **Rheobase (nA)** | 13.38 | 8.46 | 2.12 | 16 | 2.00 | 28.00 |
| **Conduction Delay (ms)** | 1.44 | 0.14 | 0.03 | 16 | 1.20 | 1.64 |
| **Action Potential Height (mv)** | 67.06 | 4.21 | 1.12 | 14 | 60.00 | 75.00 |
| **Input Conductance (S)** | 0.51 | 0.18 | 0.06 | 10 | 0.25 | 0.87 |
| **AHP Amplitude (mV)** | 1.59 | 0.81 | 0.21 | 15 | 0.70 | 3.71 |
| **AHP ½ Width (ms)** | 11.02 | 4.15 | 1.07 | 15 | 4.04 | 17.20 |
| **Twitch Force (grams)** | 2.78 | 2.09 | 0.56 | 14 | 0.50 | 6.93 |
| **Contraction Time (ms)** | 15.08 | 4.37 | 1.17 | 14 | 8.60 | 21.47 |

*Two motoneuron ap heights were excluded. In both instances spikes were blocked and not able to produce a full spike even with positive current injection.

document the expected relationship between rheobase current input and input conductance [30,38,44]. The same direct relationship emerged here in a subset of 10 motoneurons from which reliable measurements of input resistance obtained. For this sample, input conductance explained a significant amount of variance in rheobase, 64.8% (95% HDI:10.9–42.2, $\beta_1$ = 26.0). As a validation of our findings, we calculated the posterior probability density between rheobase and input conductance for a larger data base of MG motoneurons (n = 44) obtained in separate studies in our lab under identical conditions. For this larger sample, input conductance explained 64.4% of the variance (95% HDI:15.5–25.0, $\beta_1$ = 20.2) in rheobase. Fig 3B illustrates the observed data and predicted slopes for both groups of neurons derived from the probabilistic model. Near complete overlap of slopes indicates strong evidential support that the subset of 10 motoneurons did not differ in their expected relationships (i.e., slope) from neurons previously collected in our laboratory or those described in the literature [30,44].

In order to reconcile the univariate correlations between input conductance and $AIS_d$ with rheobase, we applied a generative multivariate model to understand their interdependencies. This allowed us to statistically evaluate interdependencies among parameters. Four fitted models were evaluated using Pareto smoothed importance sampling, leave-one-out cross-validation (PSIS-LOO) [50]. We focused on the expected log pointwise predictive density (ELPD) as an unbiased estimate of each model's predictive performance. This analysis yielded clear evidence that the model containing $AIS_d$ alone contained more predictive information than the model containing input conductance alone (ELPD diff = -7.2, SE = 4.8). Moreover, there was no evidence that additive (ELPD diff = -0.8, SE = 1.5) or multiplicative (ELPD diff = -2.2, SE = 1.5) models, including both parameters, improved performance (see Methods) over the $AIS_d$ model alone. This statistical evaluation of data obtained from ten motoneurons in the present study demonstrated that covariation of rheobase with $AIS_d$ did not depend on relationships with input conductance.

## AIS geometry relation to motor unit recruitment order

Close interrelationships among a constellation of physical, physiological, and biomechanical properties are a hallmark of motor units, each comprised of a motoneuron and its monosynaptically attached subset of muscle fibers. These relationships (e.g., S2) coordinate specialized properties of motor units in ways that determine usage during body movements and postures. During movements driven by slowly graded increments in the force of muscle contraction, the relationship between motor unit force and motoneuron excitability ensures progressive recruitment of motor units from the least to most forceful [51]. The present data set gives

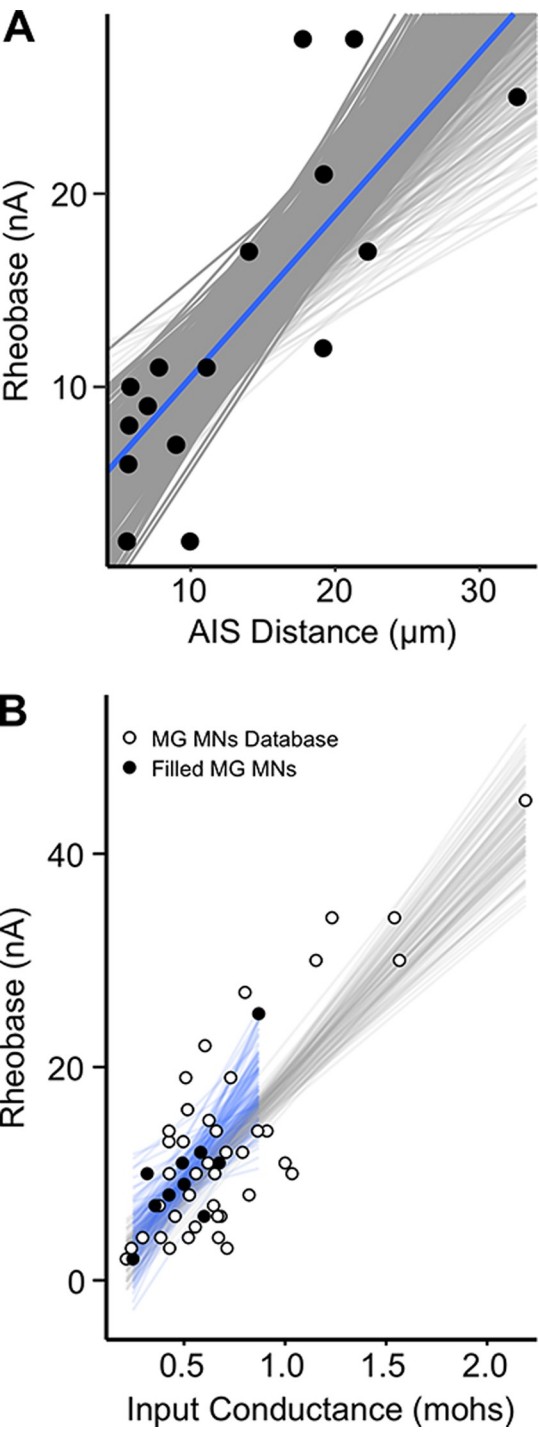

**Fig 3. $AIS_d$ covaries with rheobase.** Comparisons were performed using a Bayesian analytical approach to calculate a posterior probability distribution. Plots are based on a generative model conditioned on previous reports and the current data set. Each grey line represents a single trial from 4,000 generative samples and each black dot is an observed data point (n = 16). From the 4,000 samples we provide a 95% high density interval (HDI). The median slope from the generative sample is represented as a blue line ($β_1$). From the slopes and generative model, we compute an $R^2$ equivalent, and this is presented with an HDI and a median. (A) Rheobase: AIS distance. As rheobase increases the AIS distance increased from the motoneuron soma. Model slope: 95% HDI 0.463–1.21, $β_1$ = 0.846; $R^2$: 95% HDI 0.364–0.732, median **$R^2$ 0.631**. (B) Close similarity in rheobase and input conductance between two samples of MG motoneurons. A hierarchical Bayesian model was constructed for 44 MG motoneurons (white filled circles, grey lines) pooled from a larger MG motoneuron database produced by the Cope lab and for 10 MG Neurobiotin filled

motoneurons (black circles, blue lines) pooled from the 16 cells presented in Fig 3. Models were conditioned on previous reports and datasets from this study [44]. The positive correlation between rheobase and conductance is representative of prior reports and our small sample of 10 motoneurons fell within the expected range produced from our larger dataset. *44 motoneurons*—Model slope: 95% HDI 15.5–25.0, $\beta_1$ = 20.2; $R^2$: 95% HDI 0.522–0.728, median **$R^2$ 0.644**. *Neurobiotin filled motoneurons*—Model slope: 95% HDI 10.9–42.2, $\beta_1$ = 26.0; $R^2$: 95% HDI 0.239–0.743, median **$R^2$ 0.648**.

rough representation of this relationship and incorporates $AIS_d$ such that the most excitable, low force motor units tend to possess shorter $ASI_d$ (Fig 4). Statistically significant correlations were also found with $AIS_l$ (S2 Fig), which tended to be relatively longer for motor units that produced less force. Present data are insufficient to assess the potential mechanistic or functional meaning of correlations with AIS geometry. However, these correlations together with motor pool distinctions presented above suggest that AIS geometry may prove useful as a physical marker for roughly distinguishing motoneurons by their functional properties.

## Discussion

Our study objective was to determine whether natural variation in AIS geometry covaries with excitability among spinal motoneurons in normal mature rats. Given the dependence of this relationship on activity in other neuron types, we expected its expression by spinal motoneurons, which normally exhibit wide-ranging differences in intrinsic excitability and corresponding variation in activity patterns regularly utilized by the central nervous system to control body movement. We found that $AIS_d$ and excitability measured together from individual motoneurons were inversely correlated such that $AIS_d$ was relatively shorter longer for more readily recruitable motoneurons. In the following discussion, we conclude that while not causally related with other parameters measured here, AIS geometry is coordinated together with a variety of properties that collectively match the specialized properties of motor units to their recruitability.

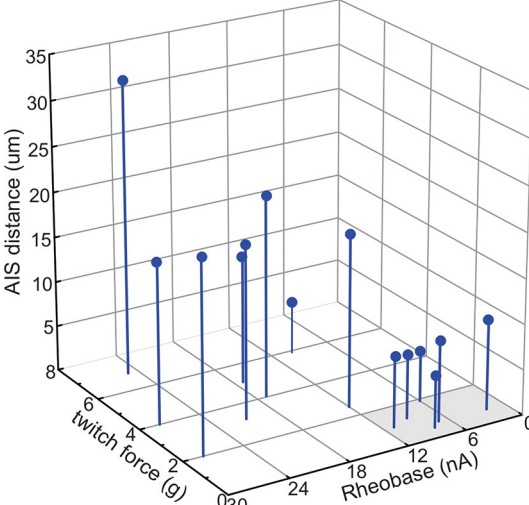

**Fig 4. AIS geometry relationship to markers of motor unit recruitment order.** 3D plot shows motoneurons with relatively lower rheobase and twitch force (grey highlight), which are typically recruited early during muscle contraction also exhibited the shortest $AIS_d$.

## Motoneuron excitability and AIS geometry

Present findings follow earlier experimental results showing that neuron excitability covaries with AIS geometry. There is precedence from other neuron types for the relationship observed in our sample of motoneurons, wherein excitability declined as $AIS_d$ increased [4,6,10,14,52]. However, the opposite relationship, excitability increasing together with $AIS_d$, has also been reported experimentally [9,14] including for motoneurons [17]. In apparent compensation for reduced motor activity in rats treated with botulinum toxin, motoneuron excitability and $AIS_d$, each measured in separate experiments, both increased [17]. Still other studies of genetically-induced motoneuron disease in mice revealed changes only in $AIS_l$ but no detected change in $AIS_d$, excitability, or firing behavior [18,19]. While the reasons for disparate findings among motoneuron studies are not evident, we suspect some influence from confounding variables introduced by pathological conditions [20–25]. In addition, relationships may be misrepresented when AIS geometry and excitability are not measured together from individual motoneurons. Whatever their origins, these inconsistent findings support current understanding that neither AIS distance nor length *per se* is a predominant determinant of excitability [2,15,53].

Excitability measured by rheobase is strongly influenced by input conductance, which represents membrane and geometrical properties of the neuron's soma-dendritic compartment. Our study of motoneurons corroborates the correlation between rheobase and conductance expected from biophysical principles and demonstrated in earlier studies (30, 44, 45). It was not surprising then to find that both rheobase and input conductance covaried with $AIS_d$. In attempt to distinguish the origins of covariation among these properties, we applied statistical modeling, which demonstrated for the present data set that $AIS_d$ rheobase had a relationship with $AIS_d$ that was not dependent on input conductance. While requiring incisive biological data for validation, this finding aligns with earlier evidence that input conductance plays a limited role in distributing the range of excitability among motoneurons [30]. These observations lead us to the provisional conclusion that the correlation between rheobase and $AIS_d$ reflects factors acting on motoneuron voltage threshold.

We have no plausible explanation for how $AIS_d$ alone might directly cause the covariation with motoneuron excitability. Biophysical modeling demonstrates that moving the AIS further away from somatic capacitance decreases voltage threshold by a few millivolts [15], in contrast with our finding that rheobase increased with $AIS_d$. It is unreasonable that the narrow range of $AIS_d$ values measured here (~20 μm) was sufficient to impact electrotonic decay, since theoretical studies suggest modest attenuation over this distance [16] and direct estimates of soma-axon voltage attenuation in pyramidal cells suggest effective length constants > 400 microns [54,55]. If not by a deterministic effect, then by what association was $AIS_d$ so remarkably predictive of motoneuron rheobase? We propose that $AIS_d$ is tightly correlated with other factors directly responsible for establishing the range of rheobase. These factors could include variation in the properties of different types of voltage-gated ion channels and their distribution within the AIS. For example, heterogeneity in the density, distribution, and phosphorylation state of Nav1.1 and 1.6 channels within the motoneuron AIS fine-tune current threshold and will therefore impact the correlation with $AIS_d$ [26,56–59]. Furthermore, simulated data suggest that expression of low-threshold voltage-gated potassium channels, such as those from the Kv1 and Kv7 family, may provide a resting hyperpolarizing current at the axon that results in higher rheobase as a function of distance from the soma by increasing outward current [26,52,60,61].

## Motor unit recruitment and AIS geometry

The recruitment of motor units under direction of motoneurons typically occurs in rank order during a wide variety of movements driven by numerous sources of synaptic drive. The

biophysical basis for systematic differences in motoneuron recruitability remains an active point of discussion [62]. Whatever the causal factors may be, it is well established that motor unit recruitment order occurs in relation to a broad set of correlated properties for both motoneurons and the subset muscle fibers they control. The linkage to recruitment order is founded on motor unit properties that accurately predict order, notably motor unit contraction force [28]. Through its correlation with motor unit force, corroborated in the present data set, rheobase aligns with the rank ordered recruitability of motoneurons. Similarly, $AIS_d$ mirrors recruitment order by covarying with rheobase and motor unit force and, therefore, may prove useful as a physical marker of the intrinsic recruitability of motoneurons.

## Methods

All procedures were performed using adult Wistar rats (M/F, 250–350 g; Charles River Laboratories, Wilmington, MA) as approved by the Georgia Institute of Technology Institutional Animal Care and Use Committee.

### Experimental design

Rats were divided into two groups. In one group, retrograde labeling was used to identify multiple motoneurons belonging to either the MG or the SOL motor pool. These motoneurons were measured for various morphological parameters, including $AIS_l$ and $AIS_d$ (see Fig 1). In the second group, both morphological and electrophysiological measures were obtained from individual MG motoneurons isolated by intracellular penetration with glass micropipettes. During all *in vivo* experimental procedures, including terminal and survival surgeries and experimental data collection, rats were deeply anesthetized (verified by the absence of withdrawal reflex) by isoflurane inhalation (5% induction, 1–3% maintenance in 100% $O_2$). At the end of experiments, rats were euthanized via an overdose of isoflurane inhalation (5%) transcardially perfused with chilled vascular rinse (0.01 M phosphate buffered saline with heparin) followed by a fixative solution (2% paraformaldehyde in 0.1 M phosphate buffer, pH 7.4). Spinal cords were removed and post-fixed for one hour then stored in 30% sucrose at 4˚C.

### Collecting and processing tissue samples from designated motor pools

Rats (n = 5) were anesthetized and surgically prepared for a brief survival surgery for the purpose of retrograde labeling of designated motor pools. The left MG and SOL muscles were exposed and injected with 30–40 μl of 0.1% Cholera Toxin Subunit B (CTB) conjugated to an Alexa-Fluor 647 or 488 (Invitrogen, Thermo Fisher Scientific, Waltham, MA). Spinal cords were collected (as described above) 72 hours post-injections.

Spinal cords were cut into 30–50 μm thick cross-sections using a Leica cryostat or microtome (Leica Microsystems, Buffalo Grove, IL) and mounted on glass microscope slides (Superfrost Plus,Fisher Scientific). Slides were incubated for 7–10 mins in cold acetone to reveal ankyrin-G (AnkG) epitopes [63]. Sections were then washed in 0.01M phosphate buffered saline with 0.3% triton (PBS-T) and incubated in blocking solution (10% normal goat serum mixed in PBS-T) for 1 hr. Serum was aspirated and replaced with a primary antibody solution containing mouse anti-AnkG (IgG2a, 1:400, Neuromab, RRID:AB_10673030), and sections were then incubated overnight at room temperature (RT) with gentle agitation. Immunoreactive regions were revealed with a mouse-specific secondary antibody raised in goat conjugated to a Cy3 anti-mouse IgG, Fcy subclass 2a specific secondary antibody (1:100, Jackson ImmunoResearch, RRID:AB_2338695) mixed in PBS-T. The sections were incubated in the secondary antibody mixture for 2 hours at RT with gentle agitation. Slides were then washed in PBS, mounted, and coverslipped with Vectashield.

## Collecting and processing both in vivo electrophysiology and tissue samples from individual motoneurons

Rats (n = 8) were prepared for *in vivo* study in terminal experiments as described previously [47,64]. Briefly, each rat was deeply anesthetized and vitals were continuously monitored including heart rate (300–500 beats/min), oxygen saturation (>90%), end-tidal $CO_2$ (2–5%), respiration rate (40–60 breaths/min), and core body temperature (37–38˚C). Next, surgical procedures were used to expose the spinal cord (L4-S1) and the MG muscle and nerve in the left hindlimb. All other peripheral nerves in the left hindlimb were crushed. Finally, each rat was secured in a stereotaxic frame configured to support recording and stimulation devices applied to exposed tissues bathed in warm mineral oil.

Individual MG motoneurons were studied via intracellular penetration within the spinal cord by glass microelectrodes coupled to an electrometer (Axoclamp). Glass micropipettes were filled with 10% Neurobiotin (Vector Laboratories, Burlingame, CA, USA) in 0.1 M Tris-OH and 1.0 M potassium acetate and had electrical resistances ranging between ∼5–10MΩ. Motoneurons were selected for study when antidromic action potentials evoked by electrical stimulation of the MG nerve and tested repeatedly during recording, exceeded 60mV in amplitude. Next, a series of biophysical properties were recorded from motoneurons having stable membrane potentials (examples in Fig 2). Voltage responses to intracellular current injection (square-pulses repeated at 1pps) were recorded to obtain conventional measures of the motoneuron's biophysical properties [38,46]. Rheobase current, referred to as rheobase from this point forward, was our designated measure of excitability and was recorded as the first among progressively incrementing current pulse amplitudes (50ms duration) to produce an action potential (Fig 2B). Input resistance was obtained from steady-state voltage responses averaged over multiple trials of 1 or 3nA hyperpolarizing current pulses (50ms) (Fig 2C) and was calculated and as the average resistance of the two values, each calculated as voltage response divided by current strength. All other text refers to the inverse of input resistance, i.e., input conductance for its common usage. Inadequate bridge balance prevented input conductance measurement for some motoneurons. Afterhyperpolarization (ahp) and muscle twitch were measured from action potentials elicited by suprathreshold current pulses (0.5ms duration) (Fig 2C and 2E). Action potentials elicited in the motoneuron during the ahp test evoked isometric motor unit twitch contractions, which were measured by a force transducer attached to the MG muscle tendon. Finally, current injection through the micropipette (5nA square pulses, 1ms duration delivered continuously at 2 Hz for 5mins) was used to fill the motoneuron with Neurobiotin (Fig 2).

Terminal experiments concluded with rat perfusion and extraction of lumbosacral spinal segments for processing and sectioning as described above. Spinal cord sections we incubated with streptavidin conjugated to an Alexa Fluor 488 (1:50; Invitrogen, RRID:AB_2315383) mixed with the secondary antibody solution for purposes of identifying Neurobiotin-filled motoneurons. Sections were also processed for AnkG immunoreactivity as described for the retrograde labeling studying.

All recorded data (electrode current and membrane voltage together with muscle force) were digitized (20kHz; Cambridge Electronic Design Power 1401), stored and later analyzed with Spike2 software.

## Image analysis

Sections containing motoneurons labeled retrogradely with CTB or injected intracellularly with Neurobiotin were imaged at high magnification using confocal microscopy (Zeiss LSM 700). Image stacks (0.5μm steps) were captured with a 63X oil immersion objective (N.A 1.4) at 0.5 digital zoom.

## Morphological analysis of motoneurons neurons identified by retrograde labeling

Image stacks of retrogradely labeled motoneurons with clear AnkG labeling were analyzed using Imaris (Bitplane, Zurich, Switzerland). Confocal image stacks were uploaded to measure the soma max cross-sectional diameter and AIS metrics using the polygon measurement tool. $AIS_d$ was measured from the axon hillock to the proximal end of AnkG immunoreactivity (Fig 1A–1C). The axon hillock was determined from a single confocal image in which the soma and the largest portion of the proximal axon were both in the same focal plane (Fig 1B1). An angle was created between the soma and axon, the apex was designated as the hillock (Fig 1B2). This was the starting point for measuring distance. The end point for the distance measurement, i.e., start of the AIS, was designated by the appearance of AnkG in a single optical plane (Fig 1B3). $AIS_l$ was then measured from the proximal to the distal ends of AnkG immunoreactivity (Fig 1 region between two red arrows, B3).

## Morphological analysis of motoneurons examined electrophysiologically

Motoneurons filled with Neurobiotin and immunolabeled with AnkG were analyzed using Neurolucida (Microbrightfield, Williston, VT). Motoneuron cell bodies were reconstructed in 3D through a series of contours traced in each optical plane. $AIS_d$ and $AIS_l$ were traced in 3D following their tortuosity through each optical plane to accurately measure both AIS distance and length. These reconstructions were also used to measure motoneuron soma surface and volume. All morphological measures were performed blind in regard to biophysical properties including rheobase.

## Statistical analysis and Bayesian modeling

Mixed-effects statistical models are because they have the power to reduce type I error rates. However, implementation of these models in a traditional frequentist framework relies on maximum likelihood estimation, which generally requires large sample sizes for model convergence and to mitigate type II errors. For small sample sizes, such as those typical *in vivo* electrophysiological studies of single neurons, mixed-effects models implemented in a Bayesian framework can overcome convergence issues, will more accurately reflect the uncertainty in effects that are based on small sample data, and are more robust to guard against the over-interpretation of unlikely results [65].

Bayesian data analytic approaches were used to empirically derive the full joint posterior probability distribution P(θ|data) of all relevant parameters (θ). analytic techniques were chosen because they present noteworthy advantages over frequentist analysis, e.g., *t*-test and ANOVA. Most notably, frequentist inference on the parameter of interest, θ, is made indirectly as it calculates P(data|θ) and not P(θ|data), as Bayesian inference does. Bayesian analytics do not require making assumptions that frequentist approaches necessitate (e.g., normally distributed data, heteroscedasticity, multiple-test correction) [66]. There is no need to generate sampling distributions, e.g., $t$, $F$, $x^2$, and determine whether the probability that unobserved data generated from the null distribution would be more extreme than the observed data i.e., formal definition of *p*-values. In addition to their ambiguity, *p* values provide an incoherent framework for reallocation of evidential support because of their high sensitivity to stopping and testing intentions [66,67].

Bayesian parameter estimation derives the entire joint posterior distribution of all parameters simultaneously and provides the ability to examine individual parameters, such as the magnitude of motor impairment across time, without the need to correct for multiple tests on

the data. In other words, there is no need for *p* values and *p* value-based confidence intervals (e.g., 95%CI). Instead, Bayesian inference and decision making provides a simpler and more intuitive interpretation of the evidence. By directly examining the posterior distributions one can see which parameter values and ranges are most credible. Further, unlike frequentists approaches, that can only reject or fail to reject a null hypothesis, by reallocating credibility across the parameter space, Bayesian inference generates evidence and uncertainty in that evidence to support <u>or</u> refute the null <u>or</u> alternative hypothesis [68]. Further, data from previously published studies can be used explicitly integrated into parameter estimation by introducing them in the prior or hyper-prior distributions depending on the hierarchical structure of the model [69,70]. For more detailed insights into Bayesian analytic approaches, we direct readers to one of many excellent reviews, e.g., [67,70,71].

Our models describe uncertainty in the response variable, e.g., rheobase, AIS distance, *y*, conditional on unknown parameters $\theta$ (e.g., regression coefficients) and predictors (e.g., biophysical and morphological parameters or motor pool membership), *x*, as well as the *a priori* uncertainty about these parameters and predictors [72]. Bayes theorem describes the proportional relationship ($\propto$) between our prior knowledge about the parameters (before observing the data) and our posterior beliefs about the parameters (after observing the data) as [72]:

$$p(\theta|y, x) \propto p(y|\theta, x) * p(\theta|x)$$

The first probability on the right-hand side of the equation is the likelihood—the joint probability of the data for all possible $\theta$ values given the observed predictors *x* [72]. The second term on the right is the prior which describes uncertainty in $\theta$ before observing data *y*. Finally, the left-hand side is the posterior—the joint probability distribution of all parameters $\theta$ after we observed data *y* [72]. It serves as a compromise between the likelihood and the prior and describes the chance of all parameter values conditional on the probability model. Evaluation of the population level AIS proximity (μm) distribution revealed a positive continuous variable, that is peaked, and has non-normally distributed heavy tails (overdispersion), so we fit a bespoke shifted log-normal regression model (Stan and rstan) with motor pool membership as the only population-level predictor to always positive predictions for the response variable as expected. For all remaining modeling analyses, we used R's *rstanarm* package, specifically the *stan_glm* function, to construct a hierarchical Bayesian model [73]. Prior to modeling, parameters were standardized.

Direct examination of the PPD enabled intuitive statistical judgments regarding the strength of the evidence, thus reallocating credibility across answers to our central questions calculating the slopes ($\beta_1$) and correlation coefficients ($R^2$). From the PPD, we derived the Bayesian equivalent of $R^2$. As defined in the standard regression model notation of:

$$y_j \sim N(u_i, \sigma) \qquad \text{Eq 1}$$

$$\text{where } u_j = X\beta$$

the $R^2$ is formulated as [69]:

$$R^2 = \frac{\sigma_f^2}{\sigma_f^2 + \sigma_e^2} \qquad \text{Eq 2}$$

$$\text{where } \sigma_f^2 = var(\mu), (u_j = X\beta), \text{ and for Gaussian models } \sigma_e^2 = var(y - \mu)$$

Marginally informative priors were applied to model parameters and variance components such that inferences were driven predominantly by the experimental data to explicitly answer our central questions. Prior specifications were based on results from previous preliminary data and published studies by our group [74] and others [30,44].

All models were fit using Hamiltonian Markov Chain Monte Carlo sampling to compute credible parameter values (θ), e.g., means, standard deviations, regression coefficients, effect sizes. Each model was run with four independent chains for 400 warm-up and 4,000 sampling steps. Steps to perform model evaluation and validation have been extensively described in our previous work [73–75]. Briefly, for all parameters, the number of effective samples was >2000, convergence was assessed and assumed to have reached the stationary distribution by ensuring that the Gelman–Rubin shrinkage statistic for all reported parameters was <1.05. Results are presented as posterior means and 95%. Bayesian credible intervals—specifically, highest posterior density intervals, which denote the 95% most plausible values of the parameter being estimated. Briefly, trace plots were examined and indicated clear stationarity and good mixing, and numerical checks of sampling quality indicated convergence (i.e., $R^\wedge = 1.0 R^\wedge = 1.0$, Monte Carlo standard error $= 0.0$, and effective sample size $> 2000$).

## Supporting information

**S1 Fig. Observed and Bayesian distributions of AIS$_d$ between MG and SOL retrogradely motoneurons. A)** Distribution of the observed AIS distance between the MG (n = 65) and SOL (n = 82) motoneurons. Black vertical line inside the plot represents the median value. MG: avg. 12.50 ± 5.95 (s.d.), SOL: 7.11 ± 2.65 (95% HDI—SOL: 6.58–7.84, MG: 11.0–14.4). **B)** Bayesian posterior predictive modeling distributions for AIS$_d$ in MG and SOL motor pools conditioned on the observed data in A). **C)** Average and **D)** standard deviation derived from shifted log-normal regression model (see Methods).
(TIF)

**S2 Fig. A pairwise comparison of morphology and biophysical properties from filled motoneurons (n = 10–16 motoneurons).** *Upper left to the bottom right diagonal*: Histogram plots showing distribution of data from motoneurons. Fitted line to the distribution of data is in red. *Left of histogram*: Scatter plots for all comparisons. Each dot represents a single data point. The open white circles represent comparisons that do not reach significance. All black circles represent significant correlations. *Right of the histograms*: Each value listed in the matrix is an r value computed from a Pearson correlation. The number of red asterisks refer to significance level corrected for multiple comparisons (*p<0.05, **p<0.01, ***p<0.001).
(TIF)

## Acknowledgments

We wish to acknowledge the core facilities at the Parker H. Petit Institute for Bioengineering and Bioscience at the Georgia Institute of Technology for the use of their shared equipment, services, and expertise. The authors would also like to thank Ms. Emily Pfahl for assisting with AIS measurements and Dr. Tom Hamm for providing intellectual feedback on the data.

## Author Contributions

**Conceptualization:** Travis M. Rotterman, Darío I. Carrasco, Randall K. Powers, Timothy C. Cope.

**Data curation:** Travis M. Rotterman, Darío I. Carrasco, Paul Nardelli, Timothy C. Cope.

**Formal analysis:** Darío I. Carrasco, Stephen N. Housley.

**Investigation:** Travis M. Rotterman.

**Methodology:** Travis M. Rotterman.

**Visualization:** Stephen N. Housley, Timothy C. Cope.

**Writing – original draft:** Travis M. Rotterman, Stephen N. Housley, Timothy C. Cope.

**Writing – review & editing:** Travis M. Rotterman, Paul Nardelli, Randall K. Powers, Timothy C. Cope.

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
