## [Decision Letter · Decision Letter 0]

6 Oct 2021

PONE-D-21-28122Axon initial segment geometry in relation to motoneuron recruitabilityPLOS ONE

Dear Dr. Rotterman,

Thank you for submitting your manuscript to PLOS ONE. After careful consideration, we feel that the study is very strong but does not yet completely meet PLOS ONE’s publication criteria as it currently stands. Therefore, we invite you to submit a revised version of the manuscript that addresses the points raised during the review process.

 The study represents very nice work showing the contribution of AIS distance to both natural variability in cellular excitability and in recruitment order. Please address the minor revisions that are requested by your reviewers and resubmit the manuscript for editorial assessment.

We look forward to receiving your revised manuscript.

Kind regards,

Peter Wenner

Academic Editor

PLOS ONE

Journal Requirements:

https://journals.plos.org/plosone/s/file?

Reviewers' comments:

Reviewer's Responses to Questions

**Comments to the Author**

1. Is the manuscript technically sound, and do the data support the conclusions?

Reviewer #1: Yes

Reviewer #2: Yes

2. Has the statistical analysis been performed appropriately and rigorously? 

Reviewer #1: Yes

Reviewer #2: Yes

3. Have the authors made all data underlying the findings in their manuscript fully available?

Reviewer #1: Yes

Reviewer #2: Yes

4. Is the manuscript presented in an intelligible fashion and written in standard English?

Reviewer #1: Yes

Reviewer #2: Yes

5. Review Comments to the Author

Reviewer #1: The authors reported the geometry of the axon initial segments of motor neurons in motor pools with different activity profiles, namely medial gastrocnemius and the soleus of rats. Using retrograde labeling of the motor neuron pools and visualization of axon initial segment (AIS) with Ankyrin G immunohistochemistry, in combination with in vivo electrophysiology in rats for evaluation of neuronal excitability, they demonstrated the correlation rheobase, as a measure of excitability, and distance from soma to AIS. However, they could not account for the relationship between their findings and the 30-fold range of rheobase for recruitment order of motor units.

Although they discussed and commented on the limitation of their conclusions, they may refer to the possible interpretation in more detail. For instance, they suggested the variation of voltage-gated ion channels in the AIS. Such kind of discussion based on literature will be greatly helpful for the broad audience of the journal.

Reviewer #2: This is a highly interesting study and very carefully done study. The data are presented clearly and the statistical analyses are both effective and elegant. There are a number of interesting results, e.g. the lack of relation between conductance and AIS distance. I have only a couple very minor comments.

Line 265: I am not sure this sentence makes sense: “Although not included in the linear regression presented in Fig 3A, values for the two motoneurons for which axons originated from dendrites.”

Line 364: I understand why the authors conclude that AIS distance is not a major driver of excitability - but I did wonder about its impact on spike voltage threshold. On the other hand, fairly direct information about voltage threshold is implicit in the relation between conductance and, e.g., rheobase current. So perhaps the relation between voltage threshold and AIS distance is already accounted for in the authors extensive (and elegant) statistical analyses?

6. PLOS authors have the option to publish the peer review history of their article (what does this mean?). If published, this will include your full peer review and any attached files.

Reviewer #1: No

Reviewer #2: No

---

## [Author Response · Author response to Decision Letter 0]

22 Oct 2021

Below we provide responses indicating how we edited the paper to address each individual point raised by the academic editor and both reviewers below: 

Reviewer 1: 

• We have elaborated on possible interpretations of our data including variability in spatial distribution and density of voltage-gated channels at the AIS within the discussion section. We also included mechanistic explanations and have provided both primary and review articles to support our claims. We agree with the reviewer that this will prove beneficial to the broad audience of PLOS One. 

Reviewer 2: 

• In reference to line 265: We thank you for catching this. This was an oversight on our part. Our intent is was to show two additional data points on the graph to display two filled motoneurons that had axons originating from the dendrites though the values of these two cells were not actually included in the statistical analysis. However, they exceeded the limits of the plot. The sentence has been fixed to reflect the current figure. 

• In reference to line 364: This is an excellent point and we have elaborated on this in the discussion however threshold was not included in our final analysis due to experimental variables that impacted our ability to collect a complete and comprehensive dataset. A common occurrence while using 10% Neurobiotin is an increase in electrode resistivity and rectification during current injection which was necessary to unblock some action potentials. This made it challenging to measure voltage-threshold accurately and rigorously in our sample.

---

## [Editor Report · Decision Letter 1]

29 Oct 2021

Axon initial segment geometry in relation to motoneuron excitability

PONE-D-21-28122R1

Dear Dr. Rotterman,

We’re pleased to inform you that your manuscript has been judged scientifically suitable for publication and will be formally accepted for publication once it meets all outstanding technical requirements.

Kind regards,

Peter Wenner

Academic Editor

PLOS ONE

Additional Editor Comments (optional):

Congratulations, nicely done!
---

## [Editor Report · Acceptance letter]

4 Nov 2021

PONE-D-21-28122R1 

Axon initial segment geometry in relation to motoneuron excitability 

Dear Dr. Rotterman:

I'm pleased to inform you that your manuscript has been deemed suitable for publication in PLOS ONE. Congratulations! Your manuscript is now with our production department. 

Kind regards, 

on behalf of

Dr. Peter Wenner 

Academic Editor

PLOS ONE